# Updated Insights into the T Cell-Mediated Immune Response against SARS-CoV-2: A Step towards Efficient and Reliable Vaccines

**DOI:** 10.3390/vaccines11010101

**Published:** 2023-01-01

**Authors:** Manish Dhawan, Ali A. Rabaan, Mahmoud M. Al Fawarah, Souad A. Almuthree, Roua A. Alsubki, Amal H. Alfaraj, Mutaib M. Mashraqi, Saleh A. Alshamrani, Wesam A. Abduljabbar, Ameen S. S. Alwashmi, Fatimah Al Ibrahim, Abdulmonem A. Alsaleh, Faryal Khamis, Jameela Alsalman, Manish Sharma, Talha Bin Emran

**Affiliations:** 1Department of Microbiology, Punjab Agricultural University, Ludhiana 141004, Punjab, India; 2Trafford College, Altrincham, Manchester WA14 5PQ, UK; 3Molecular Diagnostic Laboratory, Johns Hopkins Aramco Healthcare, Dhahran 31311, Saudi Arabia; 4College of Medicine, Alfaisal University, Riyadh 11533, Saudi Arabia; 5Department of Public Health and Nutrition, The University of Haripur, Haripur 22610, Pakistan; 6Microbiology Laboratory, Johns Hopkins Aramco Healthcare, Dhahran 31311, Saudi Arabia; 7Department of Infectious Disease, King Abdullah Medical City, Makkah 43442, Saudi Arabia; 8Department of Clinical Laboratory Sciences, College of Applied Medical Sciences, King Saud University, Riyadh 11362, Saudi Arabia; 9Pediatric Department, Abqaiq General Hospital, First Eastern Health Cluster, Abqaiq 33261, Saudi Arabia; 10Department of Clinical Laboratory Sciences, College of Applied Medical Sciences, Najran University, Najran 61441, Saudi Arabia; 11Department of Medical Laboratory Sciences, Fakeeh College for Medical Science, Jeddah 21134, Saudi Arabia; 12Department of Medical Laboratories, College of Applied Medical Sciences, Qassim University, Buraydah 51452, Saudi Arabia; 13Infectious Disease Division, Department of Internal Medicine, Dammam Medical Complex, Dammam 32245, Saudi Arabia; 14Clinical Laboratory Science Department, Mohammed Al-Mana College for Medical Sciences, Dammam 34222, Saudi Arabia; 15Infection Diseases Unit, Department of Internal Medicine, Royal Hospital, Muscat 1331, Oman; 16Infection Disease Unit, Department of Internal Medicine, Salmaniya Medical Complex, Ministry of Health, Kingdom of Bahrain, Manama 435, Bahrain; 17University Institute of Biotechnology, Department of Biotechnology, Chandigarh University, Mohali 140413, Punjab, India; 18Department of Pharmacy, BGC Trust University Bangladesh, Chittagong 4381, Bangladesh; 19Department of Pharmacy, Faculty of Allied Health Sciences, Daffodil International University, Dhaka 1207, Bangladesh

**Keywords:** COVID-19, immune response, SARS-CoV-2, T cells, T-cells’ exhaustion, Therapeutics, vaccines

## Abstract

The emergence of novel variants of SARS-CoV-2 and their abilities to evade the immune response elicited through presently available vaccination makes it essential to recognize the mechanisms through which SARS-CoV-2 interacts with the human immune response. It is essential not only to comprehend the infection mechanism of SARS-CoV-2 but also for the generation of effective and reliable vaccines against COVID-19. The effectiveness of the vaccine is supported by the adaptive immune response, which mainly consists of B and T cells, which play a critical role in deciding the prognosis of the COVID-19 disease. T cells are essential for reducing the viral load and containing the infection. A plethora of viral proteins can be recognized by T cells and provide a broad range of protection, especially amid the emergence of novel variants of SARS-CoV-2. However, the hyperactivation of the effector T cells and reduced number of lymphocytes have been found to be the key characteristics of the severe disease. Notably, excessive T cell activation may cause acute respiratory distress syndrome (ARDS) by producing unwarranted and excessive amounts of cytokines and chemokines. Nevertheless, it is still unknown how T-cell-mediated immune responses function in determining the prognosis of SARS-CoV-2 infection. Additionally, it is unknown how the functional perturbations in the T cells lead to the severe form of the disease and to reduced protection not only against SARS-CoV-2 but many other viral infections. Hence, an updated review has been developed to understand the involvement of T cells in the infection mechanism, which in turn determines the prognosis of the disease. Importantly, we have also focused on the T cells’ exhaustion under certain conditions and how these functional perturbations can be modulated for an effective immune response against SARS-CoV-2. Additionally, a range of therapeutic strategies has been discussed that can elevate the T cell-mediated immune response either directly or indirectly.

## 1. Introduction

Severe acute respiratory syndrome coronavirus 2 (SARS-CoV-2), the causative agent of the COVID-19 pandemic, has evolved into a number of variants that have shown an increased capacity to resist neutralizing antibodies (nAbs) induced either by natural infection or by the vaccination [1,2]. A recently emerged Omicron variant (B.1.1.529) is known for its high number of mutations and its altered route of infection. Additionally, the divergence of the Omicron variant into its subvariants, such as BA.2, BA.3, BA.4, BA.5, BQ.1, and BQ.1.1, has led to serious concerns regarding the available vaccines’ effectiveness. Additionally, the appearance of reinfection, even in people who have had vaccinations, raises worries about immunological escape [3,4,5]. To contain the SARS-CoV-2 infection, adaptive immune responses, especially cell-mediated immune responses, are crucial [6]. The relative significance of cellular immunity is of great concern, given the fast progression of SARS-CoV-2 and its sister variants [7]. Robust T-cell responses are linked to less severe infections. However, hyperactivation could worsen the infection. Further, T cells have long-term memory. They can respond to SARS-CoV-2 Delta and Omicron variants because they target conserved peptide epitopes [8,9]. In a viral infection, CD4+ and CD8+ T cells perform non-redundant immunologic tasks that support the innate immune system’s ability to control viral replication and nAbs’ capacity to fight infection [1,10].

T cells could be the key facilitators to control the viral infection, according to the previous findings about SARS-CoV-1 and MERS [11,12]. An integrated cell-mediated immune response is crucial for the prevention and treatment of illness [13], in addition to the significance of innate immune responses (Figure 1) [14]. Early cytotoxic CD8+ T cell development is associated with adequate viral clearance [15] and mild disease [16] and is consistent with similar kinetics for an adaptive immune response [15,17,18]. It should be noted that a portion of this response could come from bystander CD8+ T cells. By secreting cytokines like interferon-gamma, activated bystander CD8+ T cells may take part in a protective immune response. Additionally, they cause host damage via cytotoxic activity that is promoted by molecules like granzyme B and NKG2D, which activate natural killer cells. NKG2D and IL-17 have been found to be crucial markers of activated bystander-CD8+ T cells [16,19,20]. Up to 20% of COVID-19 patients have inadequate adaptive immunity, which may indicate that they might benefit from early antibody treatment despite the apparent strong cellular response in the majority of patients [7].

SARS-CoV-2 may cross-react with T cells generated by a seasonal coronavirus infection in young people, resulting in resistance to infection. Cross-reactive immune responses may be helpful due to similarities between SARS-CoV-2 and endemic seasonal cold coronaviruses (HCoVs) and a link between recent HCoV infection and a less severe course of COVID-19 in young populations. Non-SARS-CoV-2-specific T cells may reduce infection severity in children and adults. SARS-CoV-2 infection in the elderly may be linked to a decrease in cross-reactive T cells. Contradictorily, the lack of CD4+ T cell cross-reactivity between endemic beta-coronaviruses and SARS-CoV-2 suggests that these responses may have come from naive T cells rather than cross-reactive, coronavirus-specific T cells. S-proteins of variants of concern (VOCs) like Omicron and Delta have undergone significant changes. CD4+ T cell clones couldn’t recognize 7 of 17 S-protein epitopes. CD4+ T cells cannot recognize mutated epitopes. Such findings suggest T cell cross-reactivity towards VOCs is impaired. Scarce data makes it hard to draw such conclusions. Ongoing genomic monitoring is necessary to spot novel changes that may circumvent CD4+ T cell immunity [21,22,23].

Recent studies have associated improved clinical outcomes in acute COVID-19 with elevated amounts of effector molecules by CD8+ T cells [12,24]. Although polyfunctionality peaks in mild illness [24], absurdly high levels of T-cell activation are related to poor prognosis of the infection [25], suggesting that extensive stimulation of the T cells may be harmful. The extreme activation of cell-mediated immune responses has been associated with the immunopathological events in acute SARS-CoV-2, which can lead to a poor prognosis of the disease. However, in contrast to symptomatic disease, which is characterized by more polarised production of inflammatory mediators, virus-specific T-cell responses in asymptomatic infection are characterized by the balanced secretion of inflammatory cytokines such as IL-10 and IL-6 [26,27]. Furthermore, regulatory T cells (Tregs), important inflammatory response regulators in addition to CD4+ and CD8+ T cells, play a crucial function in immunological tolerance and balance [28]. At this point, it is unclear how important SARS-CoV-2-specific regulatory T cells (Tregs) are to the course of the disease [29], but systemic inflammation and severe pneumonitis are the main clinical outcomes of severe COVID-19, and virus-specific T-cell responses have been demonstrated to be a factor in tissue damage in respiratory diseases [30].

In a recent investigation, SARS-CoV-2-specific T cells were found up to 6 months following different types of vaccination, with more specific CD4+ than CD8+ T cells. According to this research, those who have received vaccinations continue to have T-cell immunity against SARS-CoV-2 and its variants, such as the Omicron variant, which may help to counteract the absence of neutralizing antibodies (nAbs) and prevent or lessen the severity of COVID-19 [31,32]. However, detailed and updated information is needed to dissect the mechanism of the T-cell-mediated immune response during the infection mechanism and its role in the severity of the disease. Additionally, the impact of the emergence of variants and their escape from the T-cell-mediated immune response will be helpful in developing more reliable and efficient vaccines in the future. Hence, the purpose of the current article is to emphasize the functions of T cells in the infection process, along with looking into the duration of the cell-mediated immune response to provide protection from recurrent infection. This article also focuses on the depletion of T lymphocytes and their exhaustion.

## 2. T Cells Response and Cytokine Storm

Individuals recovering from COVID-19 were reported to have relatively lower levels of pro-inflammatory cytokines and chemokines, even throughout the symptomatic stage [31,33]. Contrarily, patients with COVID-19 who were severely infected experienced an excessive release of cytokines and chemokines, such as interleukin (IL)-1, IL-2, IL-6, IL-7, IL-8, IL-10, granulocyte-colony stimulating factor (GCSF), and tumour necrosis factor-alpha (TNF-α) [34,35,36,37]. Notably, it has been suggested that the abnormally high levels of circulating cytokines in severe COVID-19 patients have a detrimental effect on T-cell survival and/or multiplication [38,39,40,41,42]. Post-mortem investigations of COVID-19 patients revealed considerable lymphocyte mortality in lymph follicles and paracortical regions of lymph nodes, which may have been caused, among other things, by macrophage-derived IL-6 that directly promoted lymphocyte necrosis [35,43,44]. The exaggerated release of cytokines such as IL-6 has been associated with the decreased number of T cells in severely infected patients with SARS-CoV-2.

Recently, autopsy investigations of thoracic lymph nodes and spleens revealed aberrant extrafollicular TNF-α amounts, leading to the blockage of BCL-6+ T follicular helper cell differentiation along with the significant reduction in the germinal centres [45]. There is a vicious circle between a cytokine storm or exaggerated release of cytokines and the reduction in the number of T cells during acute infection of SARS-CoV-2. Hence, targeting cytokines and their prospective signalling pathways are attractive prospective therapeutic targets for COVID-19 [43]. Similarly, based on the level of host immune system activation, T cells carry out antiviral functions or lead to tissue inflammation or injury [46]. According to preliminary research, Th17-induced vascular leakage and permeability may be facilitated by the excessive release of cytokines in patients with COVID-19 [47]. Increased cytokine levels trigger inflammatory and immunological reactions that affect the onset of acute respiratory distress syndrome (ARDS). The exaggerated stimulation of effector innate immune cells like macrophages, along with stimulated CD8+, Th1, Th17, NK, and NKT cells, might cause tissue damage by targeting virus-infected cells with increased cytokines (Figure 2) [48].

## 3. The Role of T Cells in the Infection Mechanism

The most recent information shows that getting a cell-mediated immune response without seroconversion has become a key paradigm during the COVID-19 epidemic. The basic criterion for determining the previous infection is typically the presence of Abs against a pathogen, but many people who have had significant exposure to SARS-CoV-2, such as medical professionals, exhibit virus-specific cell-mediated immune responses without showing any signs of virus-specific nAbs [49,50,51]. Previously, it has been reported in individuals who had a high level of human immunodeficiency virus (HIV) exposure, and it suggests that the cellular immune system may play a role in eliminating illness before it becomes persistent [52,53].

Although a substantial body of experimental data links the presence of SARS-CoV-2-specific T cells to disease prevention, questions have also been highlighted regarding the potential involvement of T cells in pathophysiology [54,55,56,57]. Dysregulated stimulation of both innate and adaptive immune responses was linked to protracted and heightened inflammation in severely infected patients with COVID-19 [23,54,55,56]. Also, the severity of COVID-19 was linked to a type of CD4+ and CD8+ T cells that were very active [23,54]. Activated CD8+ T lymphocytes were seen in diseased lung and brain tissue in addition to the bloodstream [58,59,60,61]. Interestingly, the antigen specificity of such highly activated cells is yet unknown. The T cells found in unusual areas where disease could be detected and displaying the signs of T cell activation and depletion may just be accidental immunological processes [62].

Increased plasmablast prevalence [54] and persistent type I IFN responses [63] were linked to severe COVID-19, suggesting that immune system components other than T cells were also found in higher percentages in damaged tissues. The clearest signs of lung pathogenesis in severe COVID-19 were, in fact, inflammatory processes maintained by myeloid-lineage cells that were exacerbated by antibody-mediated absorption of the virus [58,64,65]. It is noteworthy that when lung and blood samples from patients with severe COVID-19 were studied simultaneously, greater lung T-cell frequencies positively linked with survival, but higher lung infiltrative myeloid cells negatively connected with death [60]. The development of composite immune infiltrates in the lung was also linked to longer and more persistent respiratory symptoms after COVID-19. Elevated granulocyte and myeloid cell presence were connected with lung parenchyma changes shown on imaging, while increased B and T cell counts in bronchoalveolar lavage fluid (BALF) samples were linked to a variety of pulmonary dysfunctions (Figure 2). Additionally, Increased CD8+ tissue-resident memory (Trm) cell counts were also linked to indicators of chronic, sustained tissue destruction [61].

The significance of T-cell-mediated immune response in restricting the progression of tissue inflammation has also been underlined by recent studies. Viral control without explicit lung damage may result from a coordinated sequential modulation of CD4+ T cells into a predominant Th1 phenotype consisting of IFN-gamma and IL-10 [66,67]. The significance of IL-10 generation by T cells in determining their capacity to suppress the virus while preserving the host from lung disease has been extensively shown in mouse models of viral respiratory infections [68,69]. Notably, individuals with asymptomatic SARS-CoV-2 infections also showed the same T-cell response profile [26]. Other infection models showed substantial tissue damage because Th1-polarized CD4+ T cells were unable to transform into an IL-10-producing state [1]. Hence, Th1 cells can be important cells in determining the fate of the SARS-CoV-2 infection.

The crucial functions of T cells, particularly Th1 cells, in the SARS-CoV-2 infection have been highlighted by the presence of T cells in the BALF and their relationship to inflammation [67]. The existence of a Th1 cytokine profile has been linked to the severity of the infection, as a prolonged Th1 cytokine profile has been seen in severely infected patients with COVID-19 (Figure 2) [17]. Interestingly, it has been found that during the clearance of the infection or the reduction of the inflammation in SARS-CoV-2 infected cells, a vitamin D-dependent process occurs, which inhibits the Th1 cell-mediated production of pro-inflammatory cytokines like IFN-γ and TNF-β. On the other hand, Th2 cells are induced to produce IL-10 (Figure 2) [67,70,71]. This mechanism seems to be lacking in severe COVID-19, and curiously, a vitamin D deficit has been linked to the severity of COVID-19 in an epidemiological investigation (Figure 2) [71,72].

It has been hypothesized that pathways of functional disruption in T cells may be responsible for certain parts of the protracted pathology seen in some COVID-19 convalescents, as well as the aggravated inflammatory events that define severe COVID-19 [61]. The synthesis of TGF-β, which has currently been shown to inhibit solely natural killer (NK) cell function in severe COVID-19, may also play a role in the functional disorder of T cells [73]. The severity of COVID-19 has also been linked to changes in the T-regulatory cells (Tregs) [1,74]. Hence, further studies are required to illustrate the well-defined mechanisms of the T cells in the infection process.

## 4. T Cells and Disease Severity

The T-cell count inversely correlates with disease severity, according to many studies. Having more T cells also increases IFN-gamma production, which reduces disease severity. Peripheral T-cell depletion is correlated with COVID-19 disease severity, viral-RNA positivity, and non-survival [75]. Earlier stimulation of IFN-gamma-secreting T cells may indicate a favourable prognosis [13,24,76]. These findings explain why COVID-19 convalescence is characterized by an increase in the T-cell count and clonal expansion of SARS-CoV-2-specific T cells [77] and why inadequate or delayed T-cell response activation can lead to uncontrolled viral infection, severe lung damage, increased inflammation, and increased death rates [78]. Many recent studies comparing T cell count to disease severity have limitations, making it difficult to draw conclusions. The respiratory tract may contain more SARS-CoV-2-specific T cells than peripheral blood [79]. Age correlates with COVID-19 disease severity, which may be due to low proportions of naive CD4+ and CD8+ T cells [13,79]. Lack of naive T cells and virus-induced lymphopenia may cause disorganized and delayed cell-mediated immune responses in older people [13]. Variations in peripheral blood immune cell composition may also indicate COVID-19 severity [53,54]. Reduced blood T cell counts may be due to activation-induced cell damage in lymphoid organs [54,55] and lung tissue [80]. Dysregulation of antigen-presenting cells (APCs) can also affect cell-mediated immune response or T-cell count [40,81]. All these deregulations or perturbations can lead to a poor disease prognosis.

Patients with COVID-19 have undergone significant research regarding the changes in their lymphocyte populations. Although lymphopenia has been extensively studied, particularly in cases of severe illness, the fundamental causes of this phenomenon are still unknown [81,82,83]. The diversity of the immune response in hospitalized patients with COVID-19 has been shown by flow cytometry, including high dimensional analyses of peripheral blood T lymphocytes, which has assisted in the identification of several immune signatures that have been connected to various clinical outcomes [23]. Important research on peripheral lymphocytes has shown that patients with COVID-19 contain antigen-specific T cells at different stages of their infection [84,85,86].

In addition, single-cell RNA-sequencing investigations have been carried out on circulating T cells in patients with COVID-19 [87,88], and one study revealed an increase in antigen-specific CD4+ T cells. In other infectious diseases, such as viral diseases, the expansion of antigen-specific CD8+ T cells has been demonstrated [86,87,88]. Interestingly, patients with COVID-19 reported the presence of antigen-specific CD8+ T cells in the blood. However, these antigen-specific T cells have not shown substantiation of either a quantitative temporal increment in the severe disease or proof of antigen-specific CD8+ T-cell clonal expansion in a severe infection with SARS-CoV-2 [86]. Recently, Kaneko et al. (2022) demonstrated that as COVID-19 advances, cytotoxic CD4+ T lymphocytes (CD4 + CTLs) rise in the lungs, lymph nodes, and blood. In severe COVID-19, CD4 + CTLs are substantially enlarged in the lung parenchyma. In contrast, in severe COVID-19, CD8+ T cells are not apparent, they show enhanced PD-1 expression, and there is no evident increase in the amount of Granzyme B+ CD8+ T cells in the lung parenchyma. In severe COVID-19, CD4+ CTLs may be just as likely to induce viral clearance as CD8+ T cells and may also contribute to lung inflammation. CD4+ CTLs can also lead to fibrosis in the lungs [86]. However, further studies are required to decipher the clear role of these cells in the severity of the disease.

## 5. T Cell Responses to Vaccines

Only a small amount of published information compares post-vaccination antigen-specific antibodies, B cells, CD8+ T cells, and CD4+ T cells in the same people [34,89,90,91,92]. Vaccines aim to produce nAbs, which protect the lower airway and reduce illness severity. They could also reduce the virus’ shedding period and prevent transmission. Coordinated and long-lasting CD4+ and CD8+ T cell-mediated immune responses with the right specificity, phenotype, and function could also be crucial [38,93,94]. Accelerated expansion of vaccine-induced memory cells may be needed to boost immunity and stop COVID-19’s dispersal [89]. Phase III clinical trials of the mRNA vaccine (mRNA-1273) are now being conducted, and the immunological response that leads to the T-cell-mediated immune response has been identified. This vaccine contains an encoded spike trimer that has been prefusion stabilized and is packaged in lipid nanoparticles for intramuscular injection. The spike trimer’s prefusion form is suggested as the best target for the generation of therapies to disrupt viral machinery and reduce the risk of infection. The transmission of the virus relies on the quick activation of the RBD trimer apex to enable ACE2 binding. Therefore, the prefusion state is often transient and very unstable. The drastic structural change of the S2 subunit necessary for viral-host membrane fusion occurs next. Consequently, holding the S trimer in the prefusion configuration is of great importance in the generation of the vaccine [95,96]. In a clinical trial of such S trimer containing vaccine, Th1- type immune response consisting of IFN-γ, TNF-α, and IL-2, a dearth of Th2 type immune response consisting of IL-4 and IL-5 was reported. Additionally, CD8+ T cells, demonstrating an immunological signature consistent with the protection, were also recorded, which is unlikely to be attributed to a vaccine-enhanced disease syndrome. Substantial neutralizing efficacy, resistance against mouse-adapted SARS-CoV-2 infection in the respiratory system, and the disappearance of pulmonary pathophysiology were all associated with T-cell responses [97].

Other research on rhesus macaques found Th1-biased responses to mRNA-1273, as well as an increase in antigen-dependent CD40L/CD154 and IL-21-producing peripheral T follicular helper cells (TFH). Furthermore, a minimal or undetectable Th2 immune response was seen. It is worth noting that the strong neutralizing action was linked with immediate protection against viral replication, as well as lung disease [98]. mRNA-1273 was well tolerated in a phase I human study and elicited a CD4+ Th1-type immune response defined by TNF-α, IL-2, and IFN-γ. Furthermore, a lower level of Th2-type immune response was identified with IL-4 and IL-13. Interestingly, measurable amounts of CD8+ T cells were detected in both younger and older groups following two doses of the vaccination [99,100]. It was demonstrated in a related study by Sahin et al. (2020) that two doses of the BNT162b1 vaccine induced a Th1-type immune response along with CD8+ T cells accompanied by a negligible quantity of Th2-type immune response [101]. This indicates sufficient T-cell response despite the availability of fewer epitopes in comparison to a full-length S-protein [102,103]. According to Walsh et al. (2020), a different candidate (BNT162b2) that encodes an optimized full-length S-protein like mRNA-1273 has been chosen for progression into forthcoming phase 2/3 studies due to its favourable safety, immunogenicity, and increased likelihood of producing comparable results in a variety of populations, including older adults, as a result of a greater diversity of possible T-cell epitopes [104]. Many scientists have developed third-generation vaccines and administered them through various means. Immunogenicity profiling of DNA candidates encoding various forms of S-protein in rhesus macaques identified IFN-γ-producing CD4+ and CD8+ T cells along with the release of IL-4 secretion, implying a Th1-biased response [105].

The time frame of the T-cell-mediated immune response is an essential factor that decides the effectiveness of the vaccine candidate. In a Phase 2 trial of the adenovirus-vectored vaccine, the generation of T cells was recorded on day seven, and the peak of T cells was reported on day fourteen. Interestingly, the T-cell peak was also detectable at day 56, which was the last collection point of the trial [106]. In another phase 1/2 trial, the s-protein-specific CD4+ and CD8+ T cells were reported. The peak of proliferation and increased concentrations of IFN-γ with T cells were found at day 28 after the heterologous prime-boost regimen encoding a full-length S-protein [107]. Another recent study reported the efficacy and tolerance of a single dose of Ad5nCoV vaccination in healthy persons, as well as its effectiveness in eliciting specialized antiviral T-cell and B-cell mediated immune responses [108]. Notably, a significant rise in the release of various cytokines such as IFN-γ, TNF-α, and IL2 by T cells after two weeks of the vaccination has been reported [108]. An Ad5-nCoV vaccine has been found to be a successful candidate in eliciting a sufficient immune response to provide protection against SARS-CoV-2. Two other clinical studies using the Ad5-nCoV vaccine have also been reported. Furthermore, Kim et al. (2020) suggested that, apart from the adenovirus type 5-vectored COVID-19 vaccine, clinical research should also focus on the development of dendritic cells (DCs) or artificial APCs-based vaccines via utilizing the lentiviral vector expressing synthetic SARS-CoV-2 protein [109]. Interestingly, it is important to note that mRNA vaccines have several advantages over other vaccines, including the ability to induce T-cell-mediated immune responses, characterized by amounts of IFN-gamma secreted by T cells, as well as antibody-mediated immune responses [105,110,111,112].

## 6. SARS-CoV-2 Variants and Vaccine-Induced T Cell Immune Responses

Due to its continuous evolution, SARS-CoV-2 has diverged into various lineages and sub-lineages, which have shown variable amounts of resistance towards the vaccines. The US Department of Health and Human Services now divides SARS-CoV-2 variants into four general categories: variants of interest (VOIs), variants of concern (VOCs), variants of high consequence (VOHCs), and variants under surveillance (VUMs). Five VOCs have been identified by the World Health Organization (WHO), containing variants such as alpha, beta, gamma, delta, and omicron variants. The Omicron variant became the most dominant strain of all VOCs. The Omicron variant has a significantly higher number of mutations than other VOCs like Delta etc. Significant changes to the receptor-binding domain (RBD) and N-terminal domain (NTD) of S-protein were associated with greater transmissibility and resistance to neutralising antibodies (nAbs), which is cause for concern [4,113]. The S-protein of SARS-CoV-2 interacts with the ACE2 receptor on host cells, and this close relationship between the S-protein and receptor is a crucial determinant of how transmissible SARS-CoV-2 is [2,3,4,113]. Surprisingly, only five mutations in the S-protein of the Delta variant increased disease severity and mortality. These modifications render the Delta variant a deadly strain. The Omicron variant has more mutations, but it has been hypothesized that Delta mutations are worse than Omicron mutations [113]. The Omicron variant’s mutations and alterations changed its infection route compared to the parental strain and Delta variant. The Omicron variant uses TMPRSS2 less effectively, which is crucial for plasma membrane-mediated host cell entry. The Omicron variant prefers endosomal-mediated entry into the host cell, which increases viral proliferation in the upper respiratory tract and reduces severity [4,113]. Mutated SARS-CoV-2 strains can resist T-cell-mediated immune responses generated by natural infection or vaccination. A single-point viral mutation can eliminate the single T cell clone’s protection. Single-point mutations are unlikely to eliminate cell-mediated immune responses because T-cell receptors recognise over a thousand SARS-CoV-2 epitopes [114].

A study found that 93% of CD4+ and 97% of CD8+ T cell epitopes were totally conserved in patients infected with various strains of SARS-CoV-2 [115]. Furthermore, SARS-CoV-2-specific CD4+ and CD8+ T cell responses are less dominated by S-protein epitopes, and S-protein mutation has little effect on T cell responses [81]. All these findings show that SARS-CoV-2-specific T-cell responses may retain protection against a number of variants, such as Delta and Omicron variants. Additionally, the T-cell-mediated response elicited by vaccines reduces the risk of severity and death in patients infected with SARS-CoV-2 despite the fact that all vaccines were generated using an ancestral virus [116]. This raises a very interesting question: do we actually need variant-specific vaccines? In the following discussion, we will address these presumptions. Interestingly, it has been demonstrated that SARS-CoV-2-specific T-cell responses elicited by existing vaccines sustain substantial reactivity to the Omicron variant (a highly mutated variant of SARS-CoV-2) despite the fact that the importance of T-cell immunity is underappreciated [7]. In addition, a single amino acid addition or deletion across extensive peptidomes has little impact on the responses of polyclonal memory T cells [115,116]. According to one study, people who received the primary line of vaccinations such as mRNA1273, BNT162b2, or Ad26.COV2.S had a similar median effector T-cell response against the Omicron variant, as compared to against the parental strain of SARS-CoV-2, whereas prior infection could aid in the protection of the immune response. Effector T-cell responses were still evident in patients with undetectable nAbs against the Omicron variant [117]. A study found that approximately more than 90% of people showed the S-protein specific CD4+ T cells after immunization with ChAdOx-1, Ad26.COV2.S, mRNA-1273, or BNT162b2 at 28 days. Interestingly, there were no substantial variations in their responses against the parental strain and any other VOCs, such as the Delta and Omicron variants [31]. However, it is essential to note that Omicron’s S-specific CD8+ T cells were only detected in 63% of recipients. These memory T cells were sustained for at least six months following the booster dosage [31].

Additionally, during the breakthrough of the Omicron variant, recipients of the mRNA booster vaccination exhibited noticeably greater T-cell responses to the Omicron S-protein [118]. However, only 15% of vaccine recipients showed a decrease in CD8+ T cells against the Omicron variant [119]. This decline in T cells with time also suggests the need for booster doses among the more susceptible populations [112]. Additionally, this investigation demonstrated that immunization with two doses of BNT162b2 or two doses of Ad26.COV2.S may produce CD4+ T-cell responses to the Omicron variant [113,114]. It has been demonstrated that immunization with BNT162b2 or Ad26.COV2.S can induce widespread T-cell-mediated immune responses (including both CD4+ and CD8+) against VOCs, such as the Omicron variant. These T-cell responses are also long-lasting and can persist more than eight months after immunization [120]. Other research examined the efficiency of the BNT162b2 and Ad26.COV2.S vaccines in terms of CD8+ T-cell generation. The majority of the participants reported the efficient control of the virus. Even though they had significant neutralizing antibody titres, only 4 out of 30 macaques were unable to limit viral replication with a minimal Omicron-specific CD8 T-cell response [121,122,123]. It is important to note that the benefit of a varied immune response has been shown in cell-mediated immunity, comparable to the humoral responses of the heterologous vaccination regimens. Although more than 85% of the HLA class I epitopes were unaffected by the degree of amino acid sequences, one research indicated that people who received the BNT162b2 vaccine substantially retained CD8+ T-cell identification of Omicron S-protein epitopes [123].

According to another research study, people vaccinated with BNT162b2 reported significant protection against the infection caused by the Omicron variant. Interestingly, more than 90% of the median frequencies of antigen-specific CD4+ and CD8+ T cells were reported [124]. According to Cohen et al. (2022), peripheral blood mononuclear cells from 8 donors who received the BNT162b2 vaccine showed comparable levels of immune responses against the parental and highly mutated strain of SARS-CoV-2 [118,125]. Monitoring distinct SARS-CoV-2-specific CD8+ T-cell clones revealed that long-lived, circulating memory CD8+ T cells that lasted even a year were identified by an IFN signature, and cells also exhibited CD45RA and IL-7 receptor-alpha [126]. As a result, T-cell-response-induced vaccinations may persist for a very long period against VOCs, such as the Omicron variant [92].

The T-cell immune response to the SARS-CoV-2 S-protein and nAbs was recently assessed in naive and SARS-CoV-2 previously infected participants who received the BTN162b2 vaccine [127]. Twenty-six BTN162b2-vaccinated participants who had never received the vaccine and twenty who had already contracted the SARS-CoV-2 virus were also included. To assess the cellular immune response to cytokine levels, blood samples were taken at three different time points: baseline (before vaccination), 15 days after the first dose, and 15 days after the second dose. At one and six months following the second treatment, the nAbs were measured. One month following the second dosage, those who had previously contracted SARS-CoV-2 and received the BTN162b2 vaccine had the greatest percentage of nAbs when compared to naive people. Women, however, were more likely to noticeably drop their nAbs percentages over time. Those who had previously been infected with SARS-CoV-2 had a lower CD154+ IFN-gamma+ CD4+ T-cell response after receiving the second dose of BTN162b2. In comparison to naive subjects, the response of CD8+ IFN-gamma+ TNF-alpha+ T cells to peptide stimulation was generally greater. Furthermore, it was shown that both study groups had significantly lower levels of the cytokines IP-10, IFNs, and IL-10. Additionally, the naive participants’ median fluorescence intensity (MFI) levels of IL-6, IFN-alpha, IFN-beta, and GM-CSF dramatically decreased over time. It was shown that cellular T-cell response, nAbs generation, and serum cytokine concentration could all be affected by a prior SARS-CoV-2 infection. As a result, research on T-cell immune responses is crucial for formulating recommendations for vaccination programs. Future vaccine boosts need to be carefully considered as well because they may have an impact on the T-cell response if they continue to be induced by vaccination [127].

## 7. T-Cell Exhaustion and Disease Severity

It is quite obvious that the scientific community has sufficiently emphasized the importance of T-cell-mediated adaptive immune responses for the clearance of the virus and long-term antiviral immunity. However, it is important to consider that T-cell exhaustion can significantly contribute to the exaggerated release of the cytokines, commonly known as a cytokine storm, in severely infected patients with COVID-19 [35,36,128,129]. T-cell exhaustion is a condition in which CD8+ T cells might develop metabolic malfunctions. CD8+ T cells have diminished effector capabilities in this condition, making them unable to effectively control infections. As more and more effector T cells become exhausted, many persistent infections become significantly difficult to clear [130,131]. T-cell exhaustion is a condition of T-cell dysfunction that develops following several long-term infections and malignancies. According to Sears et al. (2021), it is characterized by suboptimal effector functionality, persistent expression of inhibitory receptors, and a transcriptional state different from that of functioning effector or memory T cells [132].

Researchers have shown via fluorescence-activated cell sorting (FACS) analysis that patients with viral infections, especially those in intensive care units, had greater levels of PD-1 on both CD8+ T cells and CD4+ T cells [26,27,28,29,30,31,32,33,34]. An antagonistic cytokine called IL-10 not only inhibits T-cell growth but also has the potential to cause T-cell exhaustion. Notably, in animal models of persistent infection, inhibiting IL-10 function has been shown to effectively prevent T-cell depletion [133,134]. Furthermore, researchers showed that the patients with COVID-19 had extremely high concentrations of serum IL-10 after SARS-CoV-2 infection, together with increased concentrations of exhaustion markers like PD-1 and Tim-3 on the T cells, implying that IL-10 may be the mechanism of action. Therefore, it could be crucial to halt the T cells’ exhaustion by providing strong antiviral therapies for the recovery of vulnerable individuals [34,133,134].

Upon stimulation, T cells may exhibit significant amounts of immunological checkpoints that are suppressive, including PD-1, TIM-3, CTLA-4, and TIGIT (Figure 3) [135,136]. On the other hand, chronic antigen stimulation might cause the development of inhibitory immunological checkpoints, which can cause a condition generally called T-cell exhaustion, which has been reported in severe viral infections [136,137,138]. The upregulation of inhibitory immunological checkpoints and the decreased expression of certain genes that produce certain essential cytokines and cytolytic molecules are signs that T cells in patients with severe viral infections or high viral loads may have a disturbed phenotype [139,140,141,142]. However, in COVID-19 patients, the effects of immune checkpoint upregulation on T-cell effector function, T-cell proliferative capacity, and viral clearance have not been well understood [141,142], which must be considered an utmost priority by the scientific community.

The markers or certain highly expressed proteins in the T cells are an interesting aspect of understanding the exhaustion of T cells, which can be associated with the severity of the disease or the decline in the protection against the infection with time. In a recent study, patients with COVID-19 were reported to have increased levels of CD4+PD1+CD57+ exhausted T cells as compared to non-symptomatic individuals [143]. Interestingly, the severely infected patients with COVID-19 were reported to have CD4+ T cells secreting lower levels of IFN-γ, IL-2, and TNF-α as compared to the patients with mild symptoms (Figure 3) [140]. Zheng et al. also reported the prevalence of CD4+ T cells with low levels of IFN, IL2, and TNF throughout the infection process. Additionally, compared to the moderately infected patients, the severely infected patients showed increased CD8+ T cells expressing elevated amounts of PD1, CTLA4, TIGIT, granzyme B, and perforin [131,132,139,140]. These findings imply that SARS-CoV2 infection may cause functional CD4+ T-cell dysfunction and support aberrant activation of the CD8+ T cells. Furthermore, the increased frequency of PD1+CTLA-4+TIGIT+ in T cells of patients with severe COVID-19 infection has been associated with T-cell exhaustion (Figure 3) [139,140,141].

Furthermore, recently it was discovered that natural killer (NK) cells and CD8+ T cells of patients with COVID-19 have a higher expression of the NK cell inhibitory receptor, NKG2A, which is characterized by lower intracellular amounts of CD107, IFN-γ, IL-2, TNF-α, and granzyme B [139,140,141]. This suggests functional perturbations in the NK and CD8+ T cells of patients with COVID-19. It is important to consider that some researchers contradict the idea that certain markers, such as PD-1, are not the essential characteristics of exhausted T cells [136,144]. Additional research is necessary to evaluate if immune checkpoint upregulation is caused by T-cell exhaustion or not, even though these results show that it is present in T cells from COVID-19 patients, especially those with severe symptoms [141,142,143,144,145,146,147,148]. On the contrary side, a subsequent study found that patients with COVID-19 and healthy subjects did not show substantial differences in terms of CD8+ T-cell exhaustion. Interestingly, even severely infected patients with COVID-19 did not report significant differences in T-cell exhaustion compared to the patients with mild symptoms of COVID-19 and healthy individuals [139,140]. In this context, Mohammed et al. (2022) stated that the above findings may be caused by variations in the illness severity criteria and in the demographics of the patients who were the subject of the investigations [149].

It is significant to note that Rha et al. (2021) showed the importance of memory T-cell responses in COVID-19 convalescents, but it is unclear what ex vivo SARS-CoV-2-specific T-cell phenotypes look like. By using MHC class I multimer labelling, researchers identified CD8+ T cells specific for SARS-CoV-2 and investigated their phenotypes and activities in COVID-19 patients with acute and convalescent disease [144]. Early in the convalescent phase, multimer+ cells displayed early-developed effector-memory characteristics. In the later convalescent phase, multimer+ cells exhibited an elevation in the proportion of stem-like memory cells. When paired with MHC class I multimer labelling, cytokine secretion experiments showed that the percentage of IFN-gamma-producing cells was much lower in SARS-CoV-2-specific CD8+ T cells than in influenza A virus-specific CD8+ T cells. Conclusively, it was found that the PD1+ cells were reported to secrete significantly higher levels of IFN-gamma as compared to PD1- cells. Hence, PD-1-expressing, SARS-CoV-2-specific CD8+ T cells remain functioning and are not yet exhausted [144]. However, these contradictions can be resolved in the future with more elaborated studies.

## 8. Immunomodulatory Approaches to Overcome T-Cell Exhaustion

Lymphocytopenia is a characteristic feature of severely infected patients with SARS-CoV-2 infection; additionally, the persistent infection can lead to T-cell exhaustion. Hence, it is essential to increase the number of functional T cells, which in turn can be an important element of a successful immune response against SARS-CoV-2. [149,150]. Consequently, the abundance and activation of T lymphocytes in COVID-19 patients are crucial for a complete cure and further protection from the infection. [149,150,151].

According to a plethora of studies, a significant number of patients with COVID-19 reported having lower concentrations of lymphocytes [34,152]. However, insufficient research has been done on the approaches and techniques that can increase the functionality and decrease the exhaustion of T cells [149]. Several immunomodulatory strategies have been employed to maintain homeostasis in the dysregulated immune response of severely infected patients with COVID-19 and to increase the number of T cells. Furthermore, overcoming the excessive release of cytokines and chemokines may be a beneficial strategy to overcome T-cell exhaustion [34,36,37]. Interestingly, due to their significant roles in regulating the innate immune response, toll-like receptors (TLRs)-interacting drugs have been demonstrated to be effective in the management of SARS-CoV-2 infection. Especially, TLRs 3, 7, and 8 are crucial pattern recognition receptors (PRRs) for RNA viruses’ sensing and are implicated in the PAMP-induced signalling cascade that produces the interferons (IFNs) required for antiviral defence [153,154]. Consequently, imiquimod, a well-known anticancer drug, can be useful in the treatment of COVID-19, as it is a TLR7 agonist and stimulates both specific and nonspecific immunological responses, as well as the generation of certain cytokines [155]. However, overexpression of the inflammation response, including cytokines and chemokines, which is often found in severe COVID-19 patients, may result in a cytokine storm which in turn leads to multiple organ damage. Hence, modulating the excessive release of cytokines or chemokines may be a possible treatment strategy among severely infected patients with COVID-19 [156,157].

Additionally, the importance of IL-6 has been described by various researchers in stimulation or the initiation of the cytokine storm. Hence, blocking the IL-6 and its signalling with IL-6 receptor blockers such as Tocilizumab and sarilumab can be an effective strategy to reduce the disease’s severity. The controlled secretion of cytokines can be an effective method to reduce T cell exhaustion. [157,158]. Furthermore, TNF inhibitors such as golimumab and adalimumab have gained popularity in the treatment of severely infected patients with COVID-19 [158]. It is essential to consider the association of the JAK-STAT pathway in the initiation of the cell-mediated immune response by releasing several cytokines. However, overexcitation of JAK-STAT signalling has been blocked with the help of ruxolitinib [149,159], which has shown promising results in the management of COVID-19. However, it will be interesting to see in the future how these immunomodulatory approaches help in the reduction of T-cell exhaustion. Apart from controlling the cytokine storm to maintain immune homeostasis, immune checkpoint inhibitors (ICIs) have the potential to reverse CD8+ T cell exhaustion, which is a feature of persistent infectious diseases like COVID-19 [159,160]. Two distinct markers of exhaustion that are expressed on CD8+ T cells—PD-1 and CD39—reflect the development of the severe form of the disease. In order to restore CD8+ T cell activity, both CD39/PD-1 pathways were blocked concurrently [161]. HIV-infected individuals receiving PD-1 and CD-39 inhibitors in this respect displayed functioning T cells rather than exhausted T cells [162,163].

Recovery of the exhausted or dysfunctional immune response must be one of the fundamental goals while creating novel therapeutic regimens against acute or chronic viral infections, including COVID-19. Patients with severe COVID-19 illness often have lymphopenia. However, determining the functionality of T cells in such patients is uncertain. In previous studies, the number of functional T cells was increased by using mAbs against PD1. It was the first study that highlighted the significance of PD1 inhibition in patients with human immunodeficiency virus (HIV) infection [164]. As we have discussed in the previous section that CTLA4, PD-1, and PD-L1 are the characteristic markers of exhausted CD8+ T cells. Overexpression of such markers lowers effector T cells and prevents them from proliferating [165]. Hence inhibiting these markers or their receptors with mAbs can be an efficient way to reduce T cell exhaustion, which, in turn, increases the protection against infection and alleviate the disease’s severity (Figure 3).

In previous studies on cancer therapy, the use of anti-PD-1 and anti-PD-L1 mAbs has been seen as a significant milestone in managing cancer. This suggests that the inhibition of PD-1 and PDL-1 can be a highly effective strategy to reduce T-cell exhaustion, which, in turn, contains or manages various infectious diseases (Figure 3) [149,166]. It is very well stated by Mohammed et al. (2022) that the restoration of worn-out or exhausted T cells may be a successful tactic to combat COVID-19 [149]. Anti-PD-1 mAbs such as Nivolumab, Monalizumab, and Avdoralimab have been studied under various clinical trials to evaluate their therapeutic effectiveness and reliability in the management of COVID-19 (Table 1) [141]. According to the data obtained from preclinical models, blocking the PD-1/PD-L1 pathway can be an effective method to enhance T cells’ restoration, but this may cause an additional inflammatory response and cellular damage [167,168]. The exploitation of the inhibition of immune checkpoints such as CTLA-4 and PDL-1 could sometimes result in lethal consequences, such as myocarditis and pulmonary oedema, raises the possibility of immune-related adverse events (irAEs) and excessive stimulation of T cells and other immune cells [141,169], but also of the immunomodulatory approaches to reduce the exhaustion of T cells to maintain the efficient and effective immune response either elicited by natural infection or a potentially available vaccine.

## 9. Other Strategies to Improve T-Cells Mediated Immune Response

Recent data from clinical studies have shown the potentialities of several efficient and reliable therapeutic methods, such as antiviral drugs, mAbs, convalescent plasma therapy, etc., for the treatment of COVID-19. Despite several advancements in the development of drugs that directly target the viral agent, other therapeutic strategies have been discovered that elicit a T-cell immune response and T-cell mediated cytokines (Table 1) [141]. Additionally, these strategies focus on the enhancement of the virus-specific T-cell responses, Th1 responses, increasing T-cell counts, overcoming the exhaustion and perturbations of the T-cells, and reducing inflammation [141,181,182].

In previous studies on pathological diseases, including viral infections, Adoptive T-cell transfer (ACT) has been found as an effective strategy to increase the T-cell number and overcome T-cell exhaustion. Utilizing virus-specific T cells and transfer through ACT can be an effective approach to enhance the number of T cells to manage COVID-19. It is possible to re-establish efficient antiviral defence by expanding allogeneic or autologous viral-specific T cells [183]. These cells can be formulated in vitro and injected into patients (Table 1). To treat severely infected patients with COVID-19, SARS-CoV2-specific T cells may be extracted from the blood of recovered patients, grown to utilize proteins of SARS-CoV2, and then used. ACT has been used only sporadically in COVID-19 because of its effectiveness, complications, and logistical difficulties. Notably, because of genetic constraints (HLA class I), it is not viable to use unmatched allogeneic T cells. Additionally, the continuous sensation of in vitro-expanded T cells to obtain appropriate cell production could indeed cause functional exhaustion, as well as the transition of T cells, and can cause a cytokine storm, which can exacerbate the COVID-19-related ailment side effects. To further demonstrate the promise of this innovative cell-based therapy for the management of COVID-19, other clinical studies based on ACT, such as NCT04351659, are now being conducted [141]. In contrast, another prospective clinical study, like NCT04401410, collected the SARS-CoV-2 specific T-cells from recovered patients and employed these cells to treat the severely infected patients with COVID-19 (Table 1) [141]. Additionally, another study used SARS-CoV-2 specific T cells with IFN-gamma exosomes to treat COVID-19 patients by enhancing the antiviral Th1-type immune response [141,153,184].

Earlier studies have identified IL7 as a crucial component that can restore the number of T cells. IL-7 can significantly raise T-cell receptor repertoire diversity and raise T-cell trafficking or mobility at the infection site [184,185,186,187]. Additionally, the proliferation of naive and memory T cells can be elicited with the use of IL-7. Furthermore, the use of IL-7 can increase the circulating pool of CD4+ and CD8+ T cells [186,187]. Crucially, IL-7 may prevent T-cell exhaustion during chronic infections. Hence, considering the plethora of advantages of IL-7, many clinical trials have been registered to uncover the effectiveness of recombinant IL7 in restoring lymphocyte counts in patients with COVID-19 (Table 1) [141].

One clinical study has been filed for the delivery of low-dose recombinant IL-2 to patients with COVID-19 as an additional treatment approach to manage ARDS and severe inflammatory response by increasing and stimulating Tregs. Because IL-2 is a potent stimulus and an essential cytokine for Treg and T-effector cell survival and expansion [188,189,190], low doses of recombinant IL2 have been shown to be safe and effective in growing and stimulating Tregs in patients with autoimmune disorders [181,190,191,192]. Hence, administering modest amounts of recombinant IL-2 to severely infected patients with COVID-19 should alleviate lymphopenia and restore the natural T-cell numbers [Table 1] [141].

## 10. Methods for Large-Scale Screening of T-Cell Responses

Apart from regulating the T-cell-mediated immune response and developing techniques to boost T-cell-mediated immunity, it is critical to precisely detect the T-cell-mediated immune response. Since we all know that the cell-mediated immune response is substantially more complex than other forms of immune response, a variety of tests must be studied and designed to quantify the cell-mediated immune response, especially T-cell-mediated immunity. Enzyme-linked immunosorbent spot (ELISpot), enzyme-linked immunosorbent assay (ELISA), and enzyme-linked lectin assay (ELLA) assays are examples of straightforward tests that enable quick, scalable results with minimum labour-intensiveness [193]. Although flow cytometry is the best option for complicated questions concerning the kind and number of T-cell subsets, its high entry barriers limit its capacity to scale. Assays based on next-generation sequencing (NGS), such as T-Detect and TACTseq, are constrained by the demand for specialised personnel and tools. Similar to ELISA/ELLA assays, recently developed qPCR-based assays like qTACT and dqTACT clearly have the advantage of affordability with the additional advantage of an internal control and objective data analysis [193].

The ELISA makes use of plates encapsulated with “capture” antibodies that bind to particular cytokines and measure their concentration in sera from patients who have SARS-CoV-2 infection or in supernatants from blood or PBMCs that have been stimulated with SARS-CoV-2 peptides. ELISAs are easy to use and are offered in the form of commercial kits with detailed instructions and reagents. The use of this test was streamlined when Qiagen launched an ELISA designed exclusively for the COVID-19 study. In order to understand the colourimetric result of an ELISA, which is translated into cytokine concentration using a serially diluted standard curve of known values, a plate reader is necessary. This test also has the added benefit that serum samples may be frozen and preserved for later use or maintained for retrospective research following whole blood stimulation. The difficulty of cross-comparing data among several detection systems with reliability and the comparatively poor sensitivity and specificity in comparison to other approaches are limitations [193]. For determining the functional quantity of T lymphocytes that are specific to an antigen, the ELISpot test has been considered a standard method. ELISpot is made to measure antigen-specific T cells down to the single cell level [194,195]. On a plate that has anti-cytokine antibodies, such as anti-IFN-gamma immobilised on it, activated cells are cultured. An additional biotinylated antibody can be used to detect the cytokines that are secreted by cells that specifically recognise the antigens. These cytokines are then trapped locally by coated antibodies. A colourimetric test is used to see individually bound dots, each of which corresponds to a single antigen-specific T cell. To identify more precise T cell subsets that express various cytokines, the Fluorospot test [196] uses fluorescent antibodies that have been differently labelled [197]. 

Additionally, functional studies have been carried out using flow cytometry analysis in COVID-19 research to determine which subsets of T cells (and other immune cell types) are up- or down-regulated. Additionally, activation-induced marker (AIM) assays do not rely on prior information about the HLA type, cytokine, or epitope being studied. It is clear that a vast array of knowledge may be acquired using this approach, but complexity also brings restrictions. High levels of ability, experience, preparation, effort, and expense, including both chemicals and apparatus, are needed for flow cytometry. Because of this, only a small number of samples can be investigated, which prevents its application as a reliable population monitoring technique. In order to answer particular biological questions that will eventually help with diagnosis and patient treatment, flow cytometry is, therefore, perfect for low-throughput, research-oriented investigations [193]. Because there are numerous methods for detecting T cell subsets and the increase and decrease of T cells, the selection of the method depends on the type of study. 

For COVID-19 diagnoses and clinical treatment, the assessment of certain T-cell responses could be beneficial. The three primary SARS-CoV-2 antigens (spike, nucleocapsid, and membrane) were recently tested for IFN-gamma T-cell responses in acute and convalescent people categorised by severity, as well as in vaccinated and unvaccinated controls. Researchers also evaluated IgG against spikes and nucleocapsids. The greatest percentage of T-cell responses were induced by S-protein [198]. When compared to convalescent patients, acute patients had a lower proportion of favourable reactions, although these responses increased with hospitalisation and severity. More than 200 days after diagnosis, IFN-gamma T-cell responses were seen in some recovering individuals. After receiving the second dose of the vaccine, only fifty percent of the recipients showed an IFN-gamma T-cell response. Compared to IFN-gamma T-cell responses, IgG responses were identified in more people, and there were only weak associations between the two. However, despite the fact that IgG synthesis was absent in certain acute COVID-19 individuals, a particular T-cell response was found. They discovered that only half of the vaccinated people had an IFN-gamma T-cell response after the second dosage and that the probabilities of an IFN-gamma T-cell response against SARS-CoV-2 were low during the acute phase but might improve with time. This study indicates the importance of T-cell screening to study the dynamics of T cells in the vaccinated population [198].

## 11. Conclusions and Future Prospects

Severely infected patients with SARS-CoV-2 have shown an excessive release of pro-inflammatory cytokines along with a plethora of chemokines, commonly known as a cytokine storm. Interestingly, upcoming data suggests a direct association of exaggerated immune response with a T-cell-mediated immune response composed of mainly CD4+ and CD8+ T-cells. Additionally, lymphocytopenia and excessive activation of Th1 and Th17 type T cells can be associated with an exaggerated immune response. Recent studies have shown that the various types of T-cell subsets, such as Th1, Th2, Th3, Tregs (T regulatory cells), and Th17, are involved in determining the fate of SARS-CoV-2 infection. Furthermore, a coordinated cell immune response, along with a humoral immune response, has been found to be essential in containing the SARS-CoV-2 infection. Moreover, the severity of COVID-19 has been associated with functional CD4+ T-cell abnormalities and CD8+ T-cell fatigue. The excessive expression of certain immune checkpoints and biomarkers, such as PD-1, PDL-1, CTLA-4, etc., has been found to be responsible for the exhaustion of CD8+ cells. Interestingly, in cases of excessive viral load or chronic infection, T-cell exhaustion can be more prevalent, as per the upcoming viewpoint of scientists. Hence, overcoming the T cells’ exhaustion by blocking the above-mentioned immune checkpoints and biomarkers can be an efficient and reliable approach to alleviating the severity of the COVID-19 disease.

Furthermore, amid the continuous evolution of SARS-CoV-2, it is essential that the presently available vaccines elicit an ample amount of the T-cell-mediated immune response. Many preliminary studies on the vaccines have found that the primary dose of the COVID-19 vaccine, along with booster doses, provide enough T-cell-mediated immune response to protect not only against parental strain but also against the highly mutated strain of the SARS-CoV-2, i.e., Omicron. This raises an interesting question for the scientific community: do we really need variant-specific vaccines, or should we focus on the booster doses of the COVID-19 vaccine? Although significant advancements have been made in understanding the immune response against SARS-CoV-2, there are still several important aspects of the T-cell immune response that need to be resolved, such as the role of different kinds of cytokines in interrupting the clonal expansion of T cells and the exact mechanisms by which exaggerated activation of T cells occurs. Additionally, the significance of immunomodulatory approaches such as inhibition of PD-1 and PDL-1 needs to be clarified, which can elicit a robust cell-mediated immune response. Accuracy is needed for the comprehensive characterisation of CD4+ and CD8+ T cell immunological responses and their relevance for a marker of subsequent defence. Key concerns for the management of the outbreak include the capacity of various vaccination regimens to elicit appropriate cellular responses and how they will assist in providing resistance amid the generation of the variants of concern (VOCs) of SARS-CoV-2. SARS-CoV-2 is continuously evolving into a number of lineages, and it has been predicted that these lineages might undergo changes or variations that allow it to largely evade T-cell-mediated immune responses. While earlier infection- or vaccination-induced memory T-cell responses have so far mostly remained strong, some evidence of T-cell escape has been described. Since extensive T-cell escape might undermine population-level immunity and have important clinical and public health ramifications, it is critical to continue monitoring the variants’ ability to evade T-cell responses [199].

## Figures and Tables

**Figure 1 vaccines-11-00101-f001:**
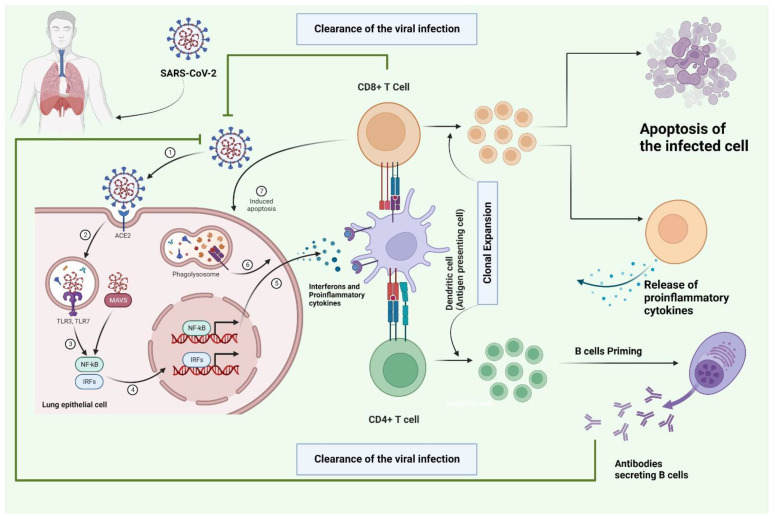
The schematic representation of the integrated adaptive and innate immune responses that result in the elimination of the viral infection. The virus enters individuals following contact with the upper respiratory tract. After virus entrance, viral replication starts by attaching to ACE2 receptors on the membrane of diverse cell types. Antigen-presenting cells (APCs), including dendritic cells and alveolar macrophages, endocytose and destroy the SARS-CoV-2 virus via a procedure termed antigen processing. Antigen segments are subsequently expressed on the cell membrane by MHC proteins, allowing T cells to recognize them. Following the interaction with T cells like CD4+ and CD8+, different types of responses occur. When CD4+ cells interact with the presented antigen on the MHC class I molecule, the activation of B-cells occurs. This will lead to the clonal expansion of CD4+ cells and B cells. After eliciting the B cells with the help of CD4+ cells, secretion of the antibodies occurs, which clears the viral infection. Upon interaction with the CD8+ T cell, clonal expansion of T cells occurs, which leads to the apoptosis of the infected cells through various mechanisms and leads to the release of other pro-inflammatory mediators.

**Figure 2 vaccines-11-00101-f002:**
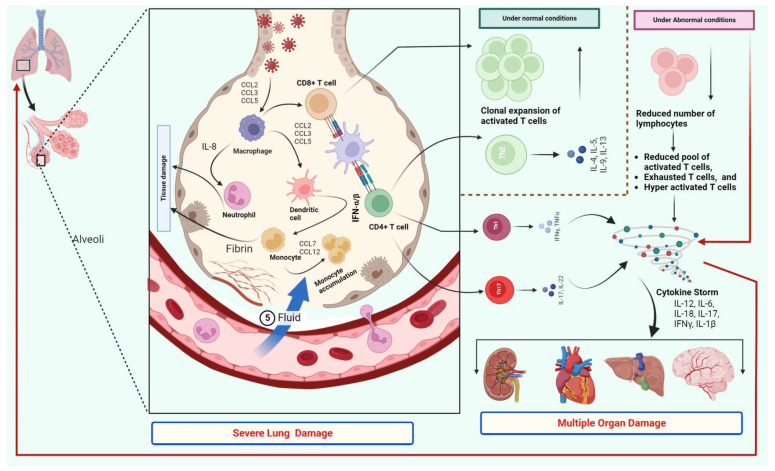
The association of the T cells with the excessive release of cytokines, disease severity, and ARDS. The expression of the ACE2 receptor on lymphocytes, particularly T cells, facilitates SARS-CoV-2 entrance into lymphocytes. During a mild infection or low viral load, the T cell number remains normal, and possibly normal functioning of the Th2 type cells will not lead to excessive inflammation and normal clonal expansion of T cells. However, under the severe form of the infection, monocyte accumulation and excessive antigen presentation lead to an increase in pro-inflammatory cytokine levels concurrently, causing T cell depletion and fatigue. Importantly excessive activation of Th1 and Th17 type T cells lead to increased levels of cytokines, importantly IL-6. Severe infection causes lymphopenia by directly affecting lymphatic organs such as the spleen and lymph nodes. The further figure represents various pathways for the excessive release of cytokines. Firstly, CD4+ T cells may be stimulated quickly into Th1 cells that secrete GM-CSF, generating CD14+CD16+ monocytes with high IL-6 levels. Secondly, an increase in the CD14+ IL-1+ monocyte subpopulation stimulates the production of IL-1. Additionally, Th17 cells generate IL-17, which recruits more monocytes, macrophages, and neutrophils while also stimulating other cytokine cascades, such as IL-1 and IL-6. This can lead to tissue damage and excessive accumulation of fluid in the lung, which in turn leads to multiple organ damage.

**Figure 3 vaccines-11-00101-f003:**
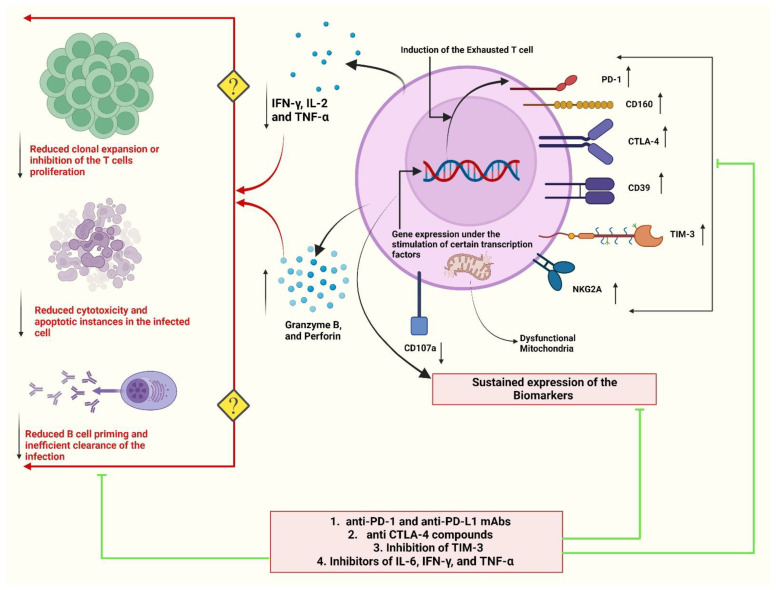
T-cell exhaustion and associated repercussions. Because of persistence antigen exposure and chronic T cell receptor (TCR) signalling, T cell exhaustion or dysregulation is common in severe viral infections. Exhausted T cells have lower proliferation and differentiation rates. Additionally, exhausted CD8+ T cells show reduced cytotoxic activities, altered gene expression, and dysfunctional metabolic activities. The dysfunctional mitochondria are also a key feature of exhausted T cells. Importantly, enhanced and persistent expression of several inhibitory receptors, notably PD-1, PDL-1, CTLA-4, CD39, and Tim-3, is a crucial factor of exhausted T cells. However, under steady-state conditions, these inhibitory receptors, also referred to as immune checkpoint proteins, are reversibly generated on activated lymphocytes and play a key role in regulating immunological balance by limiting the duration and intensity of T-cell responses. The transcription of all these receptors remains high in the exhausted T-cell state, while T-cell function diminishes, hindering effective viral elimination. Exhausted T cells release a variety of cytokines, including IFN-gamma, TNF-alpha, and IL-2, which may inhibit T-cell clonal growth and B-cell priming. As a result, all of these factors contribute to a weakened adaptive immune response. Several human mAbs have been developed to counteract these immune checkpoints to restore the T cells’ functionality even under the severe form of the disease.

**Table 1 vaccines-11-00101-t001:** Various therapeutic approaches to enhance the T cell-mediated immune response for the management of COVID-19.

Drug and Treatment Approach	ClinicalTrials.gov Identifier	Drug Candidates	Mechanism of Action	Therapeutic Benefits	Possible Side Effects	References
	NCT04268537	PD-1 blocking antibody and thymosin	Blockade of PD-1	Reversal of T-cell exhaustion; Stimulate T-cells production	Immune-related adverse events and exaggerated activation of immune cells	[141,170,171]
Inhibition of immune checkpoints	NCT04413838, NCT04356508	Nivolumab	Blockade of PD-1	Anti-PD1	Immune-related adverse events	[141,142,143,144,145,146,147,148,149,150,151,152,153,154,155,156,157,158,159,160,161,162,163,164,165,166,167,168,169,170]
	NCT04333914 NCT04333914	GNS561, Monalizumab, Avdoralimab	Inhibition of autophagy inhibitor; blocking of NKG2A; anti-C5aR	Reversal of T-cell exhaustion; Restoration of T-cell numbers; Restored effector T-cell function	uncontrolled activation of immune cells	[140,170,171,172,173]
Th1 activators	NCT04343768	Ziferon	Activation of Th1 type T cells and release of	Improved symptoms	Uncontrolled activation of immune cells	[173,174,175]
IL-6 inhibitors	NCT04320615, NCT04317092	Tocilizumab	Inhibit the binding of IL-6 with their receptors and alleviate the cytokine storm	Indirectly reduce the T cells’ exhaustion and lymphocytopenia	Efficacy and risk status of TCZ in patients at risk of other deadly infections, High cost, availability and	[170,176,177]
Th17 blockers	N.A.	Anti-IL-17, and anti-IL-22	Inhibition of the cytokines such as IL-17 and IL-22	Decreased production of IL-17, and IL-22, Activation of Th1 type cells, reduction in the viral load	N.A.	[141,176]
JAK2 inhibitor	N.A.	Fedratinib	Inhibit the excessive production of cytokines and chemokines	Reduce the inflammation	May suppress the immune response	[174,175,176]
Administration of recombinant IL-7 or IL-7 as vaccine adjuvant	NCT04407689, NCT04379076, NCT04426201	CYT107	Rearrangement of immunoglobulin (Ig) genes in immature B cell subsets regulated by IL-7. In order to maintain the diversity of T cells and the primary antibody repertoire, IL-7 also controls the T cell receptor (TCR) genes in precursor T cell subsets through the IL-7 receptor (IL-7R) signalling pathway.	Restored T-cell count and reversed lymphopenia, Enhanced TCR repertoire diversity and generation of memory CD8+ T cells, Improved trafficking of T cells to the infection site	N.A.	[141,178]
A low dose of recombinant IL-2	NCT04357444	ILT101	IL-2 maintains homeostasis in the immune response through its influence on the Tregs and effector lymphocyte responses.	Controlled stimulation of Tregs to control excessive inflammation and proliferation of Tregs and other T-cell subsets, including effector cells	Suppression of unwarranted and other inflammatory cells, including those that are necessary	[141,178,179]
Adoptive T-cell transfer (ACT)	NCT04351659	N.A.	SARS-CoV-2-reactive T cells from the patient are genetically modified ex vivo to kill virally infected cells before being reintroduced into the patient.	Overcome the T cell exhaustion. Improved specific antiviral T-cell responses against SARS-CoV-2	Risk of relapse due to several factors, including poor T-cell expansion and lack of long-term persistence after adoptive transfer	[141,180]

## Data Availability

All data are available in this manuscript.

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
