# Peer review of "Updated Insights into the T Cell-Mediated Immune Response against SARS-CoV-2: A Step towards Efficient and Reliable Vaccines"

_vaccines, 2023, doi:10.3390/vaccines11010101_

Round 1

Reviewer 1 Report

In the manuscript by Dhawan et al, T-cell mediated immune response in COVID-19 patients were reviewed. This is very timely as it discusses how vaccines boost T-cell response, protective against different variants of the virus and strategies for reversing T-cell exhaustion.

I only have minor comments and recommend publication:

1)    Please explain: NKG2D+IL-7R+ phenotype is.

2)    Please explain (line 114): SARS-CoV-2 cross-reacts with a subpopulation of T cells that have been vaccinated against seasonal coronaviruses and may help to provide therapeutic protection, especially in developing children

3)    Line 209: What is ‘plasmablast’?

4)    On Figure 1, it will be helpful to label the antigen presenting cell.

5)    What is prefusion-stabilized S trimer? Please explain.

6)    Line 393, spell out DCs as dendritic cells

7)    Line 404- sentence below is missing a verb after able to… t-cell mediated

“These vaccines are of great importance as they are able to T-cell mediated immune responses that were characterized by amounts of IFN-gamma secreted by T cells, in addition to antibody-mediated immune response.”

8)    Please provide a few statements on the differences between the variants of the virus, Omicron vs Delta vs ancestral.

9)    Typos on Figure 3.

10) Line 741, extra space was entered.

Author Response

In the manuscript by Dhawan et al, T-cell mediated immune response in COVID-19 patients were reviewed. This is very timely as it discusses how vaccines boost T-cell response, protective against different variants of the virus and strategies for reversing T-cell exhaustion.

Response: We are highly thankful for such helpful suggestions, and we believe such valuable comments will increase the credibility of the manuscript.

I only have minor comments and recommend publication:

1)    Please explain: NKG2D+IL-7R+ phenotype is.

Response: Thanks for pointing out these are the characteristic markers of the bystander CD8+ T cells, which provide non-specific protection to the viral infection. This has been explained clearly now in the revised manuscript.

2)    Please explain (line 114): SARS-CoV-2 cross-reacts with a subpopulation of T cells that have been vaccinated against seasonal coronaviruses and may help to provide therapeutic protection, especially in developing children

Response:  Thanks for pointing out, it has been explained in the revised manuscript.  

3)    Line 209: What is ‘plasmablast’?

Response: Thanks for the query. Plasmablasts are rapidly produced and short-lived effector cells of the early antibody response, whereas plasma cells are the long-lived mediators of lasting humoral immunity.

4)    On Figure 1, it will be helpful to label the antigen-presenting cell.

Response: Thanks. Done

5)    What is prefusion-stabilized S trimer? Please explain.

Response: Thanks for the comment. It has been explained in the revised manuscript.

6)    Line 393, spell out DCs as dendritic cells

Response: Thanks. Done

7)    Line 404- sentence below is missing a verb after able to… t-cell mediated

“These vaccines are of great importance as they are able to T-cell mediated immune responses that were characterized by amounts of IFN-gamma secreted by T cells, in addition to antibody-mediated immune response.”

Response: Thanks. Done

8)    Please provide a few statements on the differences between the variants of the virus, Omicron vs Delta vs ancestral.

Response: Thanks. Done

9)    Typos on Figure 3.

Response: Many Thanks. Done

10) Line 741, extra space was entered.

Response: Thanks. Done

Reviewer 2 Report

My first impression was that this a useful and comprehensive overview. It is also well organized in 10 sections. However, the paper is also a very long list of associations between immune responses and outcomes. But is in most cases the causality in this association is not clear. For instance, one would expect a presentation of the evidence that T-cells protect against disease, rather than that T cell responses develop.

In addition I have some questions:

Line 187: may be crucial in avoiding the first active infection [48-49].mitgating disease rather than prevention of infection?

Line 195: why the reference to HIV?

Line 396: The paragraph starting on this line is confusing. How can a protein integrate in the genome? Why mentioning DNA vaccines when no DNA vaccine was licensed.?

Line 409: What is lacking in section 6 is the evidence that T cells contribute to protection against covid-19.

Line 454: This is confusing participants suggests humans. The paper is about NHP.

Line 462: Refence 116 is about antibodies, not T cells.

Line 465: Define: protection against infection or disease?

Line 269: My advice is to combine section 4 and 7 in one, much shorter, section on cytokines storms and the role of T cells in cytokine storms.

Finally, in addition the evidence that T cell contribute to protection, I would have expected a discussion on simple methods for the large scale screening of T cell responses. See e.g. https://www.sciencedirect.com/science/article/pii/S2590255522000154

Author Response

My first impression was that this a useful and comprehensive overview. It is also well organized in 10 sections. However, the paper is also a very long list of associations between immune responses and outcomes. But is in most cases the causality in this association is not clear. For instance, one would expect a presentation of the evidence that T-cells protect against disease, rather than that T cell responses develop.

Response: We are highly thankful for such useful suggestions, and we believe that such valuable comments will increase the credibility of the manuscript.

In addition I have some questions:

Line 187: may be crucial in avoiding the first active infection [48-49]. mitgating disease rather than prevention of infection?

Response: We are thankful for raising concerns about such ambiguity. The changes have been made as per the query.

Line 195: why the reference to HIV?

Response: Many people who have had significant exposure to SARS-CoV-2, such as medical professionals, exhibit virus-specific cell-mediated immune response without showing any signs of virus-specific nAbs. This phenomenon has been found in HIV infection indicates a potential role for the cellular immune system in clearing infection with out the help of nAbs before it is fully established. As it is a comparable phenomenon so, the HIV related information was incorporated.

Line 396: The paragraph starting on this line is confusing. How can a protein integrate in the genome? Why mentioning DNA vaccines when no DNA vaccine was licensed.?

Response: We apologies for such typing mistakes. We have corrected the information and incorporated the relevant information also.  

Line 409: What is lacking in section 6 is the evidence that T cells contribute to protection against covid-19.

Response: We are thankful for such insightful evaluation. As this section is more about the impact of emergence of variants of SARS-CoV-2 on the T cell mediated immune response acquired by natural infection or the vaccination. So, our main focus was on impact of varinats. However, we highly agree with your concern. Hence, we  have made changes to incorporate additional information which suggests the contribution of T cells mediated immune response and how to increase it with booster doses.

Line 454: This is confusing participants suggests humans. The paper is about NHP.

Response:  Thanks, we have changed participants to the macaques, which gives a clear information to the readers.

Line 462: Refence 116 is about antibodies, not T cells.

Response: Thanks for such query, from a wide data of this research article talks about nAbs and we do understand with your concern. However, this paper also comparing the sera obtained from subjects with the CD8+ T cells. In the Discussion, the following statements have been mentioned:

[Source: 116 reference] BNT162b2 vaccination induces strong polyepitopic T cell responses, directed against multiple epitopes spanning the length of the S protein. To assess the risk of immune evasion of CD8+ T cell responses by Omicron, we investigated a set of human leukocyte antigen class I–restricted T cell epitopes from the Wuhan S protein sequence that were reported as immunogenic in the Immune Epitope Database (IEDB) (n = 244; see materials and methods). Despite the multitude of mutations in the Omicron S protein, 85.3% (n = 208) of the described class I epitopes were not affected on the amino acid sequence level, indicating that the targets of most T cell responses elicited by BNT162b2 may still be conserved in the Omicron variant (fig. S3) [116].

Hence, the information has been given with reference 116.

Line 465: Define: protection against infection or disease?

Response: Thanks, we have changed it to the infection.

Line 269: My advice is to combine section 4 and 7 in one, much shorter, section on cytokines storms and the role of T cells in cytokine storms.

Response: We are highly thankful for the suggestions. As the section 4 is particularly associating the lymphocytopenia with the disease severity. The imbalance of number of T cells such as increase and decrease of T cells was associated with the disease severity. However, section 7 is focusing on the perturbations of the T cells which leads to the T cell exhaustion. Moreover, how T cell exhaustion can lead to the severe form of the disease was explained in this section. Additionally, T cell exhaustion is not only play important role in SARS-Co-V-2 but also in various other viral infections and cancers. Hence, we think to keep these sections separate as it will be easy to link with the therapeutic strategies to overcome the T cell exhaustion in the later section.

Finally, in addition the evidence that T cell contribute to protection, I would have expected a discussion on simple methods for the large scale screening of T cell responses. See e.g. https://www.sciencedirect.com/science/article/pii/S2590255522000154

Response: We are thankful for the suggestions. A new section on the methods for large scale screening of T cell responses has been incorporated as per the suggestions.

Reviewer 3 Report

The authors aimed, in a narrative review, to emphasize the functions of T cells in the infection process, along with looking into the duration of the cell-mediated immune response to provide protection from recurrent infection. This article also focuses on the depletion of T lymphocytes and their exhaustion.

The study covers some issues that have been overlooked in other similar topics. The structure of the manuscript appears adequate and well divided in the sections. Moreover, the study is easy to follow, but some issues should be improved. Some of the comments that would improve the overall quality of the study are:

a. Authors must pay attention to the technical terms acronyms they used in the text.

b. Conclusion Section: please add some "take-home message".

Author Response

The authors aimed, in a narrative review, to emphasize the functions of T cells in the infection process, along with looking into the duration of the cell-mediated immune response to provide protection from recurrent infection. This article also focuses on the depletion of T lymphocytes and their exhaustion.

The study covers some issues that have been overlooked in other similar topics. The structure of the manuscript appears adequate and well divided in the sections. Moreover, the study is easy to follow, but some issues should be improved. Some of the comments that would improve the overall quality of the study are:

  1. Authors must pay attention to the technical terms acronyms they used in the text.

Response: We are highly thankful for the suggestions; the changes have been made accordingly, and all the terms have been checked thoroughly.

  1. Conclusion Section: please add some "take-home message".

Response: Thank you so much; the information has been incorporated.

Round 2

Reviewer 2 Report

The corrections and additions have improved the scientific value of the paper. However, the result is an even longer rambling story. Scientifically, there is nothing wrong with the paper, but I would have preferred a more concise story with  key findings on the role of T cells (protective and deletereous) relevant for a broad readership.

Author Response

The corrections and additions have improved the scientific value of the paper. However, the result is an even longer rambling story. Scientifically, there is nothing wrong with the paper, but I would have preferred a more concise story with  key findings on the role of T cells (protective and deletereous) relevant for a broad readership.

Response: 

We are highly thankful for your insightful observations, and we believe incorporating your suggestions will increase the article's readability. 

As per your suggestions, we have removed redundant and irrelevant information

We have trimmed down the article and made several changes throughout the manuscript to increase its readability. 

The manuscript was also thoroughly checked for grammatical errors

Thanks &

Best Regards